# Biological Consequences of Vanadium Effects on Formation of Reactive Oxygen Species and Lipid Peroxidation

**DOI:** 10.3390/ijms24065382

**Published:** 2023-03-11

**Authors:** Manuel Aureliano, Ana Luísa De Sousa-Coelho, Connor C. Dolan, Deborah A. Roess, Debbie C. Crans

**Affiliations:** 1Faculdade de Ciências e Tecnologia (FCT), Universidade do Algarve, 8005-139 Faro, Portugal; 2CCMar, Universidade do Algarve, 8005-139 Faro, Portugal; 3Escola Superior de Saúde, Universidade do Algarve (ESSUAlg), 8005-139 Faro, Portugal; 4Algarve Biomedical Center Research Institute (ABC-RI), 8005-139 Faro, Portugal; 5Algarve Biomedical Center (ABC), 8005-139 Faro, Portugal; 6Department of Chemistry, Colorado State University, Fort Collins, CO 80523, USA; 7Department of Biomedical Sciences, Colorado State University, Fort Collins, CO 80523, USA; 8Cellular and Molecular Biology Program, Colorado State University, Fort Collins, CO 80523, USA

**Keywords:** lipid peroxidation, vanadium, decavanadate, reactive oxygen species, oxidative stress, radicals, mitochondria, speciation

## Abstract

Lipid peroxidation (LPO), a process that affects human health, can be induced by exposure to vanadium salts and compounds. LPO is often exacerbated by oxidation stress, with some forms of vanadium providing protective effects. The LPO reaction involves the oxidation of the alkene bonds, primarily in polyunsaturated fatty acids, in a chain reaction to form radical and reactive oxygen species (ROS). LPO reactions typically affect cellular membranes through direct effects on membrane structure and function as well as impacting other cellular functions due to increases in ROS. Although LPO effects on mitochondrial function have been studied in detail, other cellular components and organelles are affected. Because vanadium salts and complexes can induce ROS formation both directly and indirectly, the study of LPO arising from increased ROS should include investigations of both processes. This is made more challenging by the range of vanadium species that exist under physiological conditions and the diverse effects of these species. Thus, complex vanadium chemistry requires speciation studies of vanadium to evaluate the direct and indirect effects of the various species that are present during vanadium exposure. Undoubtedly, speciation is important in assessing how vanadium exerts effects in biological systems and is likely the underlying cause for some of the beneficial effects reported in cancerous, diabetic, neurodegenerative conditions and other diseased tissues impacted by LPO processes. Speciation of vanadium, together with investigations of ROS and LPO, should be considered in future biological studies evaluating vanadium effects on the formation of ROS and on LPO in cells, tissues, and organisms as discussed in this review.

## 1. Introduction

Lipid peroxidation (LPO) is the autocatalyzed chain oxidation of bis-allylic bonds in the acrylic chains of fatty acids. The mechanism is well known from 30 years ago [1,2,3]. The important question is which radical starts the chain first. On the one hand, a radical is needed (oxidative stress environment) while on the other hand, LPO amplifies and contributes to changing the redox state towards oxidation (what was called oxidative stress). Reactive oxygen species (ROS) are well known to target major biomolecules, affecting several biological processes and inducing several pathologies [4]. LPO relates to aspects of oxidative stress in disease [2,5]. Oxidation of lipids (i.e., LPO), is a process occurring in cells and tissues. There appears to be a correlation between LPO products such as malondialdehyde, F_2_-isoprostanes, lipid hydroperoxides, conjugated dienes, glutathione and protein carbonyl and an imbalance between the production and accumulation of ROS leading to oxidative stress [6]. LPO effects are particularly important in organelles such as the plasma membrane and mitochondria that are surrounded by lipid-rich membranes (Figure 1) [7,8,9]. Major effects of LPO include both direct and indirect damage by ROS to key cellular functions as well as to the membranes and the endomembrane system [10]. The biological effects of LPO can include changes in membrane permeability, fluidity, permeability and integrity, as well as affecting the activity of proteins embedded in lipid environments (Figure 1) [11]. In addition, LPO has effects on cell signaling pathways that use membrane proteins dependent on specific lipid environments. As a result, LPO can be classified as a metabolic disease caused by the oxidative deterioration of lipids catalyzed by ROS [12]. The presence of metal ions such as vanadium impacts such processes in multiple ways as described in this review.

LPO has direct consequences for human health and diseases such as cancer [6]. LPO is defined as a chemical process where alkene bonds, generally in a polyunsaturated fatty acid (PUFA), are oxidized in a chain reaction to form ROS leading to a number of products [7,8]. This is similar to the toxicological definition of LPO as an oxidative chain reaction where one lipid molecule after another is oxidized to form a lipid peroxide [9]. Products of LPO can have multiple effects on cells, not only on cell components largely composed of lipids but also on other biomolecules (Figure 1). Oxidative stress is generally agreed to depend on concentrations of antioxidants and cellular exposure to free radicals (Figure 2) [10]. ROS form more frequently under conditions of oxidative stress with responses dependent on the oxidative damage potential of toxic biomolecules, drugs or other additives [7]. Alternatively, biomolecules, drugs or other additives may be protective [7]. As with other metals, vanadium has the ability to generate ROS that result in LPO and changes in biomarker levels for oxidative stress including the activity of antioxidant enzymes [13]. As discussed in the following sections, vanadium may either promote oxidative damage or play a role in antioxidant defense (Figure 2).

The primary biomarkers for LPO fall into several categories: ROS, LPO products, antioxidants and enzymes. The effects of vanadium compounds on these biomarkers have been reviewed in detail [11]. Some of these products are highly reactive, particularly the ROS derivatives, while others are more stable and can be monitored in quantitative assays reflecting the extent of LPO. It is important to note, however, that biomarkers do not always accurately describe LPO. Oxidative stress can have damaging effects on cells that are not limited to LPO, and which must be considered when analyzing data.

The extent of oxidative stress observed depends on the ratio of compounds with oxidative damage potential relative to the defense capacity of available antioxidants (Figure 2). Effects of vanadium compounds depend on whether a particular compound enhances ROS formation, in which case multiple deleterious effects on cell structure and function are observed [14,15]. Alternatively, vanadium compounds can decrease ROS formation, in which case they have protective effects on cells. For instance, it was recently described that treatment with a V complex increased the concentration of glutathione (GSH) in the visceral adipose tissue of chow-fed Wistar rats [16]. Although most studies report an increase in ROS following exposure to vanadium compounds, many have reported the protective properties of vanadium compounds [17]. This was the case for vanadyl sulfate (VOSO_4_) where administration restored the altered levels of markers of oxidative stress in the skeletal muscle of chemical-induced diabetic male Swiss albino rats, thereby potentially protecting the animals from diabetic complications [18]. As we will discuss in this review, the observed effects of vanadium compounds on the formation of ROS and LPO depend on the specific vanadium compound being studied, the biological system or specific tissue being investigated, and details of the experiment, including the duration of treatment, pH or the solution containing the vanadium compound.

The tremendous variation in vanadium effects on cells, tissues or organisms results from vanadium’s complex chemistry. Vanadium is a first-row transition metal and forms stable compounds in oxidation states IV and V under physiological conditions. Salts and coordination compounds have been investigated for effects produced under biological conditions [11]. Not only salts but also coordination complexes are found to have varying biological effects, which necessitates speciation studies together with an evaluation of biological effects. At present, several series of compounds have been studied with respect to their speciation under conditions similar to physiologic conditions, and varied biological effects have been observed [19].

Most published work studying LPO effects in biological systems has been carried out using vanadium salts. However, comparisons between the effects of monomeric vanadate (V_1_) and decavanadate (V_10_) in fish highlight the marked differences in responses seen in studies using other vanadium compounds [20,21]. This review will highlight the direct and/or indirect effects of LPO induced by V_10_, which contains 10 V atoms and 28 O atoms and forms a compact nanosized cluster that is particularly stable under slightly acidic and near physiological conditions. V_10_ has been shown to have anticancer, antiviral and antibacterial activities, among others [22,23] and, as a result, is perhaps the best-studied polyoxometalate (POM) in biology, affecting multiple biochemical and cellular processes [22,23,24]. We will also focus on the chemistry and cellular responses to vanadium compounds as they relate to LPO and oxidation stress. We and others have demonstrated different properties for various vanadium species, but an analysis that considers the formation of vanadium species over the course of an experiment has rarely been done. As an example, vanadium effects on mitochondria are of interest considering vanadium’s potential to induce both LPO directly and indirectly through the formation of ROS which then can initiate LPO. Vanadium accumulation in in vivo studies has suggested that mitochondria are a subcellular target for vanadate, especially when vanadium is administered in the form of V_10_ [25,26,27,28,29]. Because many vanadium salts and compounds have been reported to have both beneficial and toxic effects in biological systems and, in some cases, are being considered as potential therapeutic agents, it is particularly important to understand the effects of different vanadium species on LPO processes in both cell culture and animal model systems. Vanadium’s ability to form ROS through Fenton and Haber–Weiss chemistry as well as to induce LPO products directly makes it important to highlight when each mode of action occurs.

## 2. Reactive Oxygen Species and Lipid Peroxidation

### 2.1. Definition of a Reactive Oxygen Species

Radicals are reactive species with a free electron formed from the breaking of one or more bonds in the parent molecule. When the radical resides on an oxygen atom, the resulting oxygen species is very reactive. These reactive oxygen species (ROS) are fundamental to aerobic life because oxygen oxidizes both carbon- and hydrogen-rich substrates to provide energy essential for cellular functions. Important examples of oxygen-based radicals, the hydroperoxyl radical (HOO**.**), the superoxide anion radical (O_2_^−^**.**) and the hydroxyl radical (HO**.**), are products of fundamental reactions discussed below and important intermediates in lipid peroxidation.

The reduction of oxygen in the presence of a proton can lead to a number of radicals as illustrated in reactions (1)–(5). When starting with the hydroperoxyl radical shown in reaction (1), the hydroperoxyl radical can dissociate as shown in reaction (2) to form the superoxide radical anion (O_2_^−^**.**) which, upon reaction with two protons, forms hydrogen peroxide (H_2_O_2_) as shown in reaction (3). In reaction (4), the reduction of H_2_O_2_ forms one molecule each of OH^−^ and hydroxyl radical (HO**.**). HO will react with an electron and a proton to form one molecule of water as shown in reaction (5).
O_2_ + e + H^+^ → HO_2_**.**(1)
HO_2_^−^ → H^+^ + O_2_^_^(2)
O_2_^−^ + 2H^+^ + e → H_2_O_2_(3)
H_2_O_2_ + e → HO^−^ + HO**.**(4)
HO.+ e + H^+^ → H_2_O(5)

### 2.2. ROS Formed by Metal Calalysis

Reactions can occur with superoxide or H_2_O_2_ and a metal ion, a process referred to as Fenton chemistry when the metal ion is iron (Fe) [30,31,32,33]. The reaction of Fe^3+^ with superoxide is shown in reaction (6), and the disproportionation reaction catalyzed by Fe ^3+^ is shown in reaction (7). Vanadium participates in a similar reaction in which V^V^ reacts with superoxide to form a peroxyradical V^IV^-OO**.** or with NADPH, a cofactor in anabolic reactions, to form V^IV^ and NADP^+^. The disproportionation reaction (10) is catalyzed by V^IV^ and analogous to the Fe-catalyzed reaction (6). These reactions are related to the overall Haber–Weiss detoxification mechanism beginning from superoxide and shown in reaction (11) [34]. This process and the relationship between transition metals, Fenton reactions and the Haber–Weiss reaction is illustrated in Figure 3.


**Fenton reactions:**
Fe^3+^ + O_2_^−^**.**→ Fe^2+^ + O_2_(6)
Fe^2+^ + H_2_O_2_ → Fe^3+^ + HO^−^ + HO**.**(7)
V^V^ + O_2_^−^**.**→ [V^IV^OO**.**](8)
V^V^ + NADPH → V^IV^ + NADP^+^ + H^+^(9)
V^IV^ + H_2_O_2_ → V^V^ + HO^−^ + HO**.**(10)



**Haber–Weiss metal-catalyzed reaction:**
O_2_^−^**.**+ H_2_O_2_ → O_2_ + HO + HO^−^(11)


Some controversy exists in the vanadium literature resulting from reports showing that the superoxide is not kinetically competent to generate a free ROS from vanadate as shown in reactions (8) and (9) unless NADPH or phosphate is present [35]. However, recycling has been proposed with V^IV^ reacting with O_2_ to form V^V^ and the superoxide shown in reaction (12) [30,33]. Functional studies showing differences in various V compounds and their speciation are discussed in more detailed below [17,30,33].
V^IV^ + O_2_ → V^V^ + O_2_^−^**.**(12)

### 2.3. Lipid Peroxidation and Oxidized Lipid Species

LPO can be defined broadly as a chemical process in which alkene bonds, generally in a polyunsaturated fatty acid, are oxidized in a chain reaction of lipids to their respective products. At the molecular level, LPO occurs in a three-step reaction involving lipid double bonds and oxidation by ROS. This reaction results in direct and indirect damage to the cell and plays a role in several disease processes. Indirect effects of LPO are due to the reactive nature of the intermediate and final products of the peroxidative process, although, in practical terms, it is difficult to distinguish indirect effects from direct effects of LPO [36]. The effects of LPO on membranes are clearer and recognized as a common process in various pathological conditions [37]. More importantly, the continued propagation of ROS contributes to various diseases. This confirms the notion that an organism’s innate defense system is not always sufficient to protect it from damage or cell death [38,39,40].

LPO often occurs in response to oxidative stress in a reaction with three phases: initiation, propagation and termination (Figure 4A). A basic chemical description of the three-step mechanism of the LPO process is shown below. During each phase, specific lipid oxide intermediates are formed and reactions take place, as has been extensively reviewed [36,41,42,43,44].

The initiation phase is characterized by the removal of the bis-allylic hydrogen and formation of a conjugated diene system and either a pentadienyl radical (L·) or a lipid peroxyl radical (LOO·) (reactions (13) and (14) and Figure 4A). This can occur through several different mechanisms characterized as existing oxidative reactions. Oxidation can occur through reactions containing metals, pre-existing lipid hydroperoxides (see LOOHin Figure 4A), enzymes such as lipoxidase, ultraviolet light irradiation or ROS reactions. An exception is the superoxide radical which cannot initiate LPO by itself [45,46].


**Lipid peroxidation mechanism reactions**



*Initiation*
LH + HO·→ L·+ H_2_O(13)
LOOH + HOO· → LOO·+ H_2_O_2_(14)



*Propagation*
L + O_2_ → LOO·(15)
LOO + LH → LOOH + L·(16)
LOOH + M^2+^ → LO·+ HO-(17)



*Termination*
LOO·+ L → LOOL(18)
LOO·+ LOO·→ LOOOOL(19)
LOOOOL → LOH + L = O + O_2_(20)
L·+ L → L-L(21)
LOO·+ AOX → LOOH + AOX·(22)


Initiation through the formation of a free radical allows for the propagation of LPO. The propagation phase consists of a radical abstraction step that can continue indefinitely in a cyclic reaction (reactions (15)–(17)). Regeneration of the lipid radical is accompanied by the formation of a lipid peroxyl radical in reaction (15) in which the reaction of L· with molecular oxygen (O_2_) forms LOO· [42]. LOO· then reacts with a lipid through the removal of the bis-allylic hydrogen to create LOOH and regeneration of L· in reaction (16). The lipid hydroperoxide can react with a metal to form a lipid alkoxyl radical (LO·) and a hydroxide ion. This cycle is repeated until the overall reaction is terminated.

Termination of LPO propagation occurs through multiple steps and the formation of several products including, but not limited to, peroxides, aldehydes and alcohols [47]. The presence of the radical is the driving force behind the continued oxidation of lipids. Removal of the radical will cause the chain reaction to cease. There are several ways for termination to occur (Table 1, reactions (18)–(22)) and a variety of products that can result. These include the Russell peroxyl radical termination [48] to form a ketone and alcohol along with molecular oxygen, quenching of the radical via antioxidants to form lipid hydroperoxides, reactions with other lipids and lipid peroxides and the reaction of lipid hydroperoxides with metals. Termination steps and products formed are summarized in Table 1. Yin, Xu and Porter (2011) and de Groot and Noll (1987) describe the termination reactions in depth [49,50].

Lipid oxide products of LPO vary, with major diene hydroperoxide products dependent on the PUFA being oxidized. The reactions of the side products can be seen in reactions (18)–(22). LPO oxidation results in LPO products that are sterically either trans or cis and which can also vary due to the continued reaction of the peroxyl radical converting to L· [47]. Another reaction that increases variety in LPO products occurs in the oxygen cycle cyclization step in which the peroxyl radical is transferred to an adjacent carbon atom [51] (Figure 4B). This oxygen scrambling step increases product variations and can be monitored via HPLC and UV-Vis spectroscopy [47,52]. The steric variation of these hydroperoxides determines which end products are formed and whether they are subsequently oxidized to aldehydes, ketones or remained alcohols (Figure 5).

### 2.4. Effects of Vanadium Compounds and Speciation on Lipid Peroxidation

As with other toxic metals, vanadium has the ability to produce ROS resulting in LPO and changes in biomarker levels including the activity of antioxidant enzymes discussed in Section 2.6 below [13]. LPO is initiated by metal ions or metal compounds through Fenton chemistry and affects LPO reactions. The Fenton reaction produces HO· through a reaction of H_2_O_2_ with a transition metal. Vanadium, in the form of V^IV^ (vanadyl), reacts similarly with H_2_O_2_, as do other transition metals such as iron and copper, to form the HO·. This reaction initiates LPO which is followed by propagation and termination for PUFAs and lipid hydroperoxides. Due to its reactivity, the vanadium ion is often bound to ligands in complexes, with studies demonstrating anticancer or antidiabetic properties of these vanadium complexes [17,24,53,54,55].

Because vanadium’s role in LPO depends on its oxidation state and whether the vanadium is complexed with ligands, it is necessary to consider the speciation of vanadium, particularly under cellular conditions. Vanadium exists in multiple oxidation states (II-V), with oxidation states IV and V being most prevalent under physiological conditions [56,57,58], although oxidation state III may also be present but difficult to observe [59]. The properties of these different oxidation states affect the reactivity of the observed ROS. Vanadium has been reported to redox cycle between V^IV^ and V^V^, leading to the generation of ROS [17,56,58,60,61,62]. Keller and others have shown that vanadyl (V^IV^) as VO^2+^ is better at generating ROS in the form of HO· than vanadate (V^V^) [17,56,58,60,61,62]. Vanadium in oxidation states IV and V forms oxo-ions which exist as several oligomeric ions. In oxidation state IV, VO^2+^ (VO(H_2_O)_5_^2+^ and the deprotonated form VOOH^+^ (VO(H_2_O)_5_OH^+^) are the most common species considered, although it is known that these forms do not exist at neutral pH but are instead bound to proteins, DNA, lipids or other available metabolites and referred to generally as VO^2+^ or V^IV^. In oxidation V, oxovanadates are well defined, although several interconvert. The most common ions have nuclearities of 1, 2, 4, 5 and 10, each of which is likely to have different effects on lipid peroxidation. Vanadate, when its nuclearity is one (V_1_), resembles phosphate and exists in three different protonation forms, H_2_VO_4_^−^, HVO_4_^2−^ and VO_4_^3−^. These forms are readily generated when dissolving metavanadate and orthovanadate into solutions at neutral pH. Ammonium metavanadate and sodium metavanadate form V_1_ solutions and increase LPO products along with V^IV^ (as VOSO_4_) [11]. This may be due to the recycling between V^IV^ and V^V^ systems reported under physiologic conditions.

The effects of decavanadate (V_10_) on lipid peroxidation have been well studied in comparison with monomeric vanadate (V_1_) [19,25,26,27,28]. Lipid peroxidation by V_10_, unlike V_1_, requires longer times, perhaps due to the need for decomposition of V_10_ to vanadate before redox cycling and lipid peroxidation can occur. The effects of V_10_ on lipid peroxidation are also longer lasting than those of vanadate, although the mechanism is not clear. This may be due to the length of time needed for the conversion of V_10_ to V_1_, this is not consistent with other differences in the effects of these two oxovanadates. These results suggest that different LPO mechanisms are utilized by vanadate and V_10_ [19,25,26,27,28].

Similar trends in reactivity are not always observed for complexed forms of vanadium. Although metavanadate salts increase LPO products along with VOSO_4_, complexes containing both V^V^ and V^IV^ do not increase LPO products [11]. A study examining vanadium complexes with anticancer activity containing V^III^, V^IV^ or V^V^ showed that V^IV^ complexes induced higher levels of ROS necessary for LPO than did V^V^ complexes [63]. Another study investigating V^III-V^ complexes and the production of reactive oxygen and nitrogen species showed that V^IV^ produced the lowest levels of reactive oxygen and nitrogen species in colon cancer Caco-2 cells, a result attributed to V^IV^ being more inert under cellular conditions [63]. This result was also consistent with observations that V^IV^ when polymerized is slow to react as in solutions of VOSO_4_ at pH 7 [56,58,64]. Thus, if a solution of V^IV^ is polymerized before treatment of the Caco-2 cells, V^IV^ would not be available for initiation of LPO. Together, these results suggest that product formation is sensitive to the nature of the V^IV^ solutions and whether V^IV^ is polymerized [19]. They are supported by the observation that the addition of a reducing agent such as glutathione or NADPH contributes to ROS and LPO production, presumably by allowing V^V^ to cycle to V^IV^ where it initiates LPO.

In summary, vanadium can induce ROS formation in biological systems through (1) Fenton-type reactions [36], (2) bioreduction of vanadate by the action of glutathione (GSH), flavoenzymes or NAD(P)H oxidases with subsequent formation of ROS [65,66], and (3) the indirect promotion of ROS, probably by interactions with mitochondria [67]. Several hypotheses have been formulated to address the relationships between vanadate and other vanadium compounds and ROS production, including how the oxidation state and presence of ligands may impact ROS production. There remains much work to be done to investigate which species are present in vivo, the redox chemistry of the relevant vanadium compounds and their mechanisms and pathways involved in physiologic function that are affected by various vanadium species.

### 2.5. Biomarkers Associated with Lipid Peroxidation

Lipid peroxidation creates multiple products, some of which serve as biomarkers indicative of oxidative stress. The primary biomarkers for LPO fall into several categories: ROS, LPO products, antioxidants and enzymes. Some of these products, for example, ROS derivatives, are highly reactive, while others are more stable and can be monitored in quantitative assays that accurately reflect the extent of LPO. As a caveat, biomarkers may not always accurately describe LPO simply because oxidative stress can have damaging effects on cells that are not limited to LPO and should be considered when analyzing data. Regardless, biomarkers can provide important details on cell status and serve as useful reporter groups.

Because of the integral role of ROS in LPO, the production of ROS is linearly correlated with LPO formation. Flow cytometry with suitable probes has been used to measure the presence of ROS. Specific LPO products exist that reflect LPO formation and can be monitored, some of which are shown in Figure 6. As an example, malondialdehyde (MDA) is commonly used in the thiobarbituric acid (TBA) assay. Additional products that can be monitored are 4-hydroxy-2-nonenal (4-HNE), acrolein, isoprostanes and neuroprostanes [68,69]. Each of these compounds is reactive in vivo both chemically and after processing. However in vivo processing can be problematic, as is the case for MDA, which is further metabolized in vivo, making cellular metabolism an important consideration when interpreting in vivo results with this agent [70].

Biomarkers used for studying vanadium effects on LPO include MDA, ROS production, ascorbate [11] and quantification of fluorescence from *cis*-parinaric acid [27,71]. Since vanadium is known to generate ROS, monitoring ROS production offers insight into which vanadium species plays a role in ROS generation and subsequent LPO. Fluorescence decay in the presence of cis-parinaric acid (Figure 6) can be used to determine the rate of LPO in cells. The addition of peroxyl radicals to cell media containing cis-parinaric acid will cause a decrease in fluorescence emission [72]. Both V^IV^ and vanadate solutions generate ROS, although V^IV^ generates ROS more quickly and in greater amounts.

The biomarker ascorbic acid was used in two animal studies described in a review by Scibior and Kurus [11]. Mice were treated with NaVO_3_ and NH_4_VO_3_ to determine whether there was a difference between treatments using these two salts. Since these salts differ only in having a Na^+^ versus NH_4_^+^ counter ion, no major difference in LPO was expected. However, as Scibior and Kurus reported, there were large variations in LPO responses in liver tissues, as shown in Figure 7. While lower ascorbic acid (L-AA) levels were observed in the liver with NH_4_VO_3_ treatment, the L-AA content doubled with NaVO_3_ treatment. This was an unexpected result, particularly given similar responses to these salts in erythrocytes and plasma samples (Figure 7). However, these differences could be attributed to a different length of administration, the concentration of vanadate salts used, and animal age and weight, all factors that may affect L-AA content [11]. Nonetheless, it is reasonable to conclude that there are similarities between the NaVO_3_ and NH_4_VO_3_ effects on erythrocytes and in plasma but differences may be observed in the liver (Figure 7).

Enzyme activity can also be used as an LPO biomarker, as has been done using superoxide dismutase (SOD) and catalase (CAT) in studies of fish treated with vanadium [26,27,28,71]. SOD protects against the superoxide anion while CAT degrades H_2_O_2_. Both are involved in ROS metabolism and are affected by vanadium species. One study evaluating the effect of vanadate and V_10_ on the activity of SOD and CAT in cardiac tissue showed a 115% increase in SOD activity and no change in CAT activity with exposure to metavanadate and a 20% increase in SOD activity and a 55% decrease in CAT activity with exposure to V_10_ [28]. A study looking at vanadium pentoxide exposure in rabbits found a decrease in SOD (14.3%) and CAT (30.0%) activity with a 42.9–60.0% increase in LPO [73]. No study to date has compared the effects of vanadium on LPO using different biomarkers. Thus, while providing valuable details on the metabolic state of the system, there is a possibility that different biomarkers may be responding differently to various vanadium compounds.

### 2.6. The Effects of Vanadium on LPO of Proteins and Enzymes

When discussing the effects of vanadium on the LPO of proteins, it is important to consider whether an isolated protein is being studied [24] or whether the protein is studied in cells, tissues or intact animals [74]. Furthermore, the form of vanadium involved and its speciation affect the interpretation of the results [21,23,24,25,26,27,28,32,71,75]. Several common forms of V^V^ include sodium metavanadate, ammonium metavanadate, sodium orthovanadate (abbreviated SOV with a formula Na_3_VO_4_) and V_10_ which, when prepared in aqueous solutions, will form the appropriate oxovanadates as determined by solution pH [23]. In studies with isolated enzymes, one would expect similar responses regardless of the initial form of vanadium. Results from work with V^IV^ are more varied, which has been attributed to the fact that V^IV^ can polymerize to generate a slowly accessible form of V^IV^. Early studies by the Chasteen and Crans group demonstrated that buffer used experimentally could interact with V^IV^ and form a complex that prevented polymerization of V^IV^ and resulted in effects of V^IV^ in enzyme studies [64,76,77].

Vanadate is competent to reduce enzymes with, for example, thiol groups, as has been reported for protein tyrosine phosphatases [78,79]. Accordingly, one would expect these proteins to be particularly sensitive to LPO in in vivo systems and in cells. Vanadium coordination compounds may be less likely to show similar responses because the redox properties of the complex have been changed by the ligand [80,81]. Some work has been done with various vanadium compounds and isolated enzymes together with accompanying speciation studies [82]. Of particular interest when considering speciation effects is a study showing that V_10_ binding to G-actin prevents V_10_ decomposition [83]. Incorporating vanadium in a polyoxidovanadate may protect the vanadium from hydrolysis while still allowing redox chemistry. A number of studies with this class of compounds have appeared recently showing that some of these compounds are very active in vivo and have properties that suggest their effects may involve LPO [22,23,25,26,27,28].

Independent animal studies and studies in fish have allowed for a direct comparison of the effects of vanadate and V_10_, with results showing different effects on myosin, actin and P-type ATPases [22,23,24,25,26,27,28]. Only V_10_ induced the oxidation of the so-called “fast” cysteines, or exposed cysteine Cys-374, when actin was in the polymerized and active form [84]. For P-type ATPases, quercetin prevented protein cysteine oxidation induced by V_10_ [85]. For myosin, V_10_ strongly inhibited actomyosin ATPase hydrolysis and the mechanism involved in muscle contraction due to a specific interaction at the myosin backdoor [86,87]. In addition, V_10_ was found to have both direct and indirect effects on oxidative stress markers and LPO [22,23,24,25,26,27,28,71], leading to the proposed mechanisms shown in Figure 8. The redox cycling of V^V^ and direct interactions with H_2_O_2_ are likely to be both important and consistent with expected vanadium chemistry.

Vanadium compounds have also been reported to be effective initiators of signal transduction by a G protein-coupled receptor (GPCR), specifically the luteinizing hormone receptor (LHR) [88,89], and a protein tyrosine kinase, the Type I Fcε receptor [89,90]. The LHR signals in the presence of some vanadium-containing compounds as a result of vanadium compound interactions with the membrane lipid interface. Three different types of interactions that initiate cell signaling have been identified. The first class of coordination complexes was hydrophobic complexes that initiated signaling by interacting with the outside surface of the membranes, causing changes in lipid organization that indirectly drove receptor dimerization and production of a second messenger, cyclic AMP (cAMP). Evidence for this type of signaling included measuring lipid packing in the bilayer and intracellular levels of cAMP. Compounds that interact with membranes through this mechanism included VOSO_4_ [88]. The second class of compounds was vanadium coordination complexes that interacted with the cellular interface and caused changes in lipid packing. These compounds showed evidence for uptake and internalization by the cell. Vanadium compounds that were found to interact with the membrane interface and potentially be internalized included vanadate, VO(acac)_2_, and BMOV, a compound that was in clinical trials for the treatment of type 2 diabetes. In separate studies, the addition of DIDS [91] stopped the transport of these compounds across the erythrocyte membrane [67,92]. This group of compounds may also include vanadate monomer V_1_, V_10_, a polyoxovanadate with 14 V atoms (V_14_) and a polyoxovanadate with 15 V atoms (V_15_) which, because they cannot be readily removed from cells, may be internalized [93]. Finally, the third class of compounds was hydrophilic compounds that interacted only transiently with the membrane, changed lipid packing and initiated receptor signaling without penetrating the membrane. These compounds, specifically two monosubstituted V_10_s, V_9_Mo and V_9_Pt [94], could be quickly separated from cells. Although these studies provided no direct information on the impact of these compounds on LPO, membrane interactions are a feature of both vanadate and VOSO_4_ which do impact LPO.

Recently new classes of vanadium coordination complexes have been reported that are mild radical initiators and should be able to initiate LPO and induce oxidation [14]. Interestingly, they are not cytotoxic to embryonic mouse fibroblasts (NIH/3T3) [14], Cal33 cells or HeLa cells. One class of compounds is vitamin E chelate siderophores with a lipid portion and a metal ion, amphiphilic properties and hydrolytic stability [15]. The other class of compounds are vanadium complexes with a sterically hindered catechol that have low toxicity while remaining reactive, suggesting that such vanadium compounds may be active as LPO inducers and, in the case of this latter complex, demonstrate some efficacy against cancer cells [95,96].

## 3. Effects of Vanadium and LPO on Plasma Membranes, Organelles, Mitochondria and DNA

Vanadium salts and compounds can affect LPO in intact cells by enhancing ROS formation. Multiple deleterious effects of LPO include effects on enzymes involved in redox cycling of vanadium, inhibition of enzymes and proteolysis as described above in Figure 8, and damage to cellular membranes including the plasma membrane, mitochondrial membranes and membranes making up the endomembrane systems as summarized in Figure 9.

ROS and/or reactive nitrogen species (RNS)-mediated oxidative modifications in proteins [69] that produce partial inactivation or affect structure and function have been described for myosin and actin [97,98,99]. More recently, the effects of V_10_ and peroxynitrite on myosin have been compared. Peroxynitrite strongly inhibits actomyosin-ATPase activity with an IC_50_ of inhibition of 47 µM. This inhibition is due to the oxidation of the highly reactive Cys on myosin. In contrast, exposure to V_10_ induces the oxidation of a core Cys on myosin and not the highly reactive Cys together with inhibition of actomyosin-ATPase activity (IC_50_ = 2.7 µM) which is 17 times lower. These results suggest that myosin is more sensitive to oxidative modifications mediated by V_10_ than by peroxynitrite [100].

DNA effects involve changes in DNA structure which are, as a result, categorized as mutations. The general effects on the cell membrane, endomembranes and organelles including mitochondria and DNA are summarized below. When discussing the effect of vanadium on plasma membranes and cells, it is important to discuss the speciation and form of vanadium. Several commonly studied forms associated with LPO in cells are salts and include sodium metavanadate, ammonium metavanadate, vanadyl sulfate, SOV and V_10_, but as described above under Section 2.4, comparisons between studies are difficult. Much less is reported about LPO in response to vanadium coordination complexes.

### 3.1. Effects of LPO and Vanadium on the Plasma Membrane

The plasma membrane contains up to 75% phospholipids. Polyunsaturated fatty acids (PUFAs) are major lipid components in the sn2-FA position of membrane phospholipids [101,102]. LPO can break down the FAs to ketone-, alcohol- or aldehyde-containing products which affect membrane fluidity, increase membrane permeability and decrease lateral mobility of membrane proteins [103,104]. An increase in the membrane permeability changes the distribution of salts across the membrane which in turn affects the membrane potential. Increased permeability also allows undesired species to penetrate the cell membrane such as additional ROS and metals such as vanadium. Studies have shown that some LPO products change the structural conformation within the bilayer to reduce bilayer thickness and increase toxic molecule penetration of the membrane [103].

LPO in healthy cells reduces defense mechanisms protecting the cell from oxidative stress and increases the damage done by ROS. Several studies have shown that vanadate and vanadyl lower L-ascorbic acid and glutathione levels in cells, reducing overall oxidative stress in the body, decreasing organ weight and increasing LPO products [105]. Additionally, an increase in HO· concentration is observed with the addition of vanadium salts to cells, perhaps as a result of vanadium converting molecular oxygen first to H_2_O_2_ and then to HO·. Such continued imbalances in the oxidative defense mechanisms and the amount of ROS in the body will perpetuate LPO in healthy cells and cause continued damage.

Studies using vanadium in several systems including cell plasma membranes have been reported. For example, an early study investigated the interaction of vanadate with membrane preparations of the Na^+^/K^+^ ATPase using ^51^V NMR spectroscopy [106]. Interactions between vanadium and membranes showed selective interactions with V_10_, although no direct effect on LPO was reported [106]. Numerous studies with vanadate and V_10_ in membrane model systems have been carried out [85,107,108], without any evidence for reduction, and these results are consistent with the requirement for NADPH/phosphate or another reducing agent for the formation of V^IV^ leading to LPO [109].

### 3.2. The Effect of LPO and Vanadium on the Endomembrane System

Damage caused by LPO extends to organelles within the cell as well as fragmented pieces of the endoplasmic reticulum called microsomes [110]. Microsomes can be oxidized by metal ions, including vanadium, as a result of high concentrations of PUFAs in their lipid membrane [110,111,112]. The endoplasmic reticulum, part of the endomembrane system, is essential for protein synthesis and metabolism of molecules and elements such as lipids and calcium [113]. An important role of the endoplasmic reticulum is to encapsulate molecules to be transported to the Golgi apparatus [113]. For suitable vesicle formation, a mobile membrane is essential, and LPO will limit the mobility of the endoplasmic reticulum (ER) membrane components. LPO has been measured in microsomes from placenta, brain and liver cells [111,112,114]. LPO in microsomes requires vanadyl or vanadate, the latter in the presence of NADPH which is needed to reduce vanadate (V^V^) to V^IV^ so that it can initiate the LPO. The proposed mechanism is shown in Figure 10 [112].

### 3.3. The Effects of Vanadium and LPO in the Mitochondria

#### 3.3.1. LPO in the Mitochondria

Oxidative phosphorylation is a biochemical process used to meet the energy requirements of cells. Mitochondria, sites of oxidative phosphorylation, are typically found in high numbers in tissues with high energy requirements such as the heart and skeletal tissues. Any compound inhibiting oxidative phosphorylation causes dramatic, negative effects on metabolism in organs such as the heart, kidney, liver and brain. LPO increases mitochondrial membrane permeability and the permeability of the transition pore in the mitochondrial membrane as a result of the oxidation of the lipids in the membrane and an accompanying decrease in membrane thickness. This affects mitochondrial energy production which is dependent on the membrane potential and the synthesis of ATP driven by the proton gradient powering the ATP synthase. Additionally, LPO of the mitochondrial membrane can cause the release of cytochrome C, a complex necessary for electron transport [115,116]. Complex II, also known as succinate dehydrogenase, is also critical for electron transport [117] and for metabolism in the Krebs cycle [117]. The vanadium-induced decrease in complex II activity is toxic to the cells and results in cell death. Assessing various mitochondrial activities, such as oxygen consumption, ATPase activity and NADH oxidase activity, is important in monitoring the degree of mitochondrial dysfunction induced by an external, toxic agent.

#### 3.3.2. The Effect of Vanadium Salts and Complexes on the Mitochondria

Vanadium is reported to affect mitochondria and subsequent energy production, contributing to cell death [118,119]. For vanadate to generate ROS, it needs to be reduced to V^IV^ which occurs in the presence of NADPH and phosphate. The effects of different vanadium species on mitochondria have been investigated, and different responses to vanadium salts and vanadium complexes provide insight into the function of these species. LPO and both V^IV^ and V^V^ treatments [115,120] cause an increase in ROS production, with V^IV^ producing a greater increase in ROS than V^V^. LPO was measured for both types of vanadium, and although vanadyl caused more LPO than vanadate, vanadate-induced LPO could be increased with the addition of NADPH (and phosphate) [109,121], consistent with the reduction of vanadate to vanadyl to generate ROS and initiate LPO.

The total pro-oxidant activity after in vivo exposure to vanadate was evaluated through the quantitative analysis of ROS production in a study comparing the effect of two types of V^V^ vanadates, namely stable, compact V_10_ and simple vanadate, V_1_. In fish cardiac mitochondria, only the V_1_ solution caused a significant and delayed increase in the ROS production of about 198% 12 h after administration of V_1_ solution [28]. There was no effect on oxidative activity levels in V_10_-treated animals (Figure 11). In hepatic tissue [27], vanadate oligomers similarly affected the pro-oxidant activity differently than V_10_ (Figure 11). The V_10_ solution increased ROS production by about 80% during the entire exposure period (12 h), while the V_1_ solution induced a 150% increase after the first hour of exposure, registering a blockage in its pro-oxidant capacity over the 12 h of exposure with values of 40% and ~0% 6 and 12 h after exposure to vanadium, respectively [27,28] (Figure 11). The most likely interpretation for the latter result is that putative decomposition of V_10_ causes gradual, delayed exposure of the liver to a V_1_ species whose toxicity is prevented by glutathione (GSH). In fact, a similar suppression in GSH levels has been reported in the liver using the same experimental conditions for V_1_ administered in the form of metavanadate [27,28].

These effects on mitochondria have been studied using vanadium species in different oxidation states and with different ligands bound [115]. Varying responses were observed depending on the vanadium compound used. The coordination complexes VO(acac)_2_ and VOcitrate (VOcit) reacted differently than the coordination complex VOdipic with respect to the release of cytochrome C. No mitochondrial swelling was seen in the VOdipic-treated system, indicating that cytochrome C was released via a channel rather than through an increase in membrane permeability. In contrast, the two other V^IV^ complexes VO(acac)_2_ and VOcit increased membrane permeability. VOdipic had little effect on mitochondrial membrane swelling or changes in membrane potential. VOSO_4_ and VO(acac)_2_ caused swelling, eliminated the membrane potential and generated ROS. These experiments showed that the V^IV^ complexes exerted different effects depending on the ligand bound to vanadium, with the greatest effects observed for VO(acac)_2_ and the V^IV^ salt (VOSO_4_). A smaller effect was observed using NaVO_3_ and VOcit. NaVO_3_ eventually had similar effects to those of VOSO_4_, suggesting that vanadate had to be reduced to V^IV^ before becoming toxic. Effects with V^IV^ complex VOcit took longer, which was attributed to the high stability of this complex; the slow dissociation of the ligand was necessary before more pronounced effects of the complex were observed. Together, these results demonstrate that vanadium coordination with a ligand can produce subtle differences in biological activity.

#### 3.3.3. The Effect of V_10_ on the Mitochondria

As described above, vanadium can affect biological systems including mitochondria directly or through the formation of ROS, both of which can take place simultaneously [21,27,28]. In practical terms, both the direct and indirect effects of vanadium must be considered. To experimentally test the possibility that V_10_ inhibits the electron transport chain through a complex IV inhibition mechanism, the effect of V_10_ on the oxidation-reduced state of purified ferrocytochrome C was examined [122]. Neither V_10_ nor V_1_ induced changes in the oxidation-reduced state of cytochrome C. Furthermore, none of the species caused changes in cytochrome C oxidase activity in rat liver or fish heart [122,123], in agreement with previous studies [124]. These results suggest that the oxidation of reduced cytochrome C during vanadate-stimulated NADH oxidation requires the presence of vanadate and H_2_O_2_, since it occurs by the action of hydroxyl radicals formed in the vanadate/H_2_O_2_ mixture.

After excluding the notion that mitochondrial depolarization and inhibition of oxygen consumption promoted by V_10_ resulted from the inhibition of cytochrome oxidase, the effects of V_10_ on the oxidation-reduced state of the mitochondrial cytochrome b were considered [122]. Changes recorded between 500 and 550 nm induced by V_10_ in mammalian liver mitochondria indicated that V_10_, but not V_1_, altered the oxidation-reduced state of cytochrome b, suggesting that mitochondrial complex III was a target of V_10_ and that V_10_ inhibited the respiratory chain of hepatic mitochondria in a manner similar to antimycin-A, a complex III specific inhibitor. Thus, the reduction of cytochrome b is a consequence of the blockade induced by V_10_ in the respiratory chain [122].

Vanadate affects mitochondrial respiration by altering electron transfer between complexes III and IV [125]. Several studies have associated vanadium toxicity with its ability to induce ROS formation, probably through interactions with mitochondrial oxidative–reductive centers [31,67,125]. Vanadate blocked the transfer of electrons in the respiratory chain between cytochromes b1 and C, causing inhibition of succinate oxidation and oxidation of substrates associated with NADH [126]. Vanadate ions have also been implicated in the inhibition of mitochondrial succinate dehydrogenase as well as ATP-dependent succinyl-CoA synthetase from rat brain mitochondria [127]. Studies involving V_10_ and other oxovanadate species have demonstrated that V_10_, but not another oxovanadate, stimulates NADH oxidation in the erythrocyte plasma membrane and in rat liver microsomes [124,128,129], induces cytochrome c reduction [130], exhibits α-adrenergic agonist activity in aortic rings in rats [131] and is reduced by NADP^+^-specific isocitrate dehydrogenase (IDH) [130].

The effects of V_10_ and V_1_ on mitochondrial function in preparations of rat liver and fish heart were compared with respect to depolarization of the mitochondrial membrane [26,28,29]. V_10_ strongly depolarized the mitochondrial membrane potential in rat liver mitochondria [122] and fish heart mitochondria [123], compared to V_1_ species. An attempt was made to determine the site and nature of V_10_ effects on the electron transport chain by examining oxidation–reduction changes in the cytochromes of the mitochondrial respiratory chain. The effects promoted by the V_10_ and V_1_ on the potential of the mitochondrial membrane in the presence of a physiological concentration of GSH (5 mM) were further analyzed. A 10 min exposure to increasing concentrations of vanadate increased mitochondrial depolarization. V_10_ induced mitochondrial depolarization in rat liver mitochondria at very low concentrations, with an IC_50_ value of 38.7 ± 10.2 nM, while 5.4 ± 2.5 μM monomeric vanadate was required to induce 50% depolarization in rat liver mitochondria [122]. Likewise, V_10_ induced depolarization of fish cardiac mitochondria at very low concentrations, with an IC_50_ of 196 nM, while 55 µM V_1_ induced a 50% depolarization [123]. Because the V_10_ species was found to affect mitochondrial membrane repolarization (IC_50_ ~1 μM V_10_, i.e., 10 μM total vanadium) and since mitochondrial membrane hyperpolarization has been described as an early mitochondrial response during apoptotic events [132,133], these results suggest that V_10_ should be explored as an anti-apoptotic agent. Interestingly, the antibiotic cyclosporine A (CsA) did not protect mitochondria from depolarization induced by the V_10_, suggesting that, at least in isolated mitochondria, V_10_ depolarization of the mitochondrial membrane is not due to the opening of membrane transition pore inhibitor (MMTP) [134,135].

The effects of V_10_ and V_1_ on the mitochondrial function in preparations of rat liver and fish heart were compared with respect to oxygen consumption by the mitochondrial membrane [26,28,29]. V_10_ appeared to be about 100-fold more effective than V_1_ in inhibiting oxygen consumption in hepatic mitochondria, as indicated by the IC_50_ values, 98.5 ± 5.1 nM for V_10_ and 9.7 ± 1.4 μM for V_1_ [122]. Likewise, V_10_ inhibited oxygen consumption in fish cardiac mitochondria more strongly than V_1_, with an IC_50_ of approximately 400 nM for V_10_, while a 60 times higher value was determined for V_1_ (23 μM) [123]. Both vanadate solutions inhibited mitochondrial respiration without uncoupling mitochondria; there was no effect on the respiratory control ratio (RCR), which in coupled mitochondria from rat liver, breathing in pyruvate and malate, was 5.1 ± 0.1 and 5.0 ± 0.1, respectively, in the absence or presence of both V_10_ and V_1_. These results are in agreement with previous studies [29], where there was no uncoupling of oxidative phosphorylation but a slight increase in the ADP/O ratio in rat liver mitochondria, due to an inhibition of adenylate kinase activity by V_10_ (1 mM).

#### 3.3.4. Effects of V_10_ on LPO in Mitochondria

Although the indirect effects of vanadium on LPO resulting from the reactive nature of the intermediate and final products in the peroxidative process are difficult to distinguish from the direct effects of vanadium [36], indirect effects of V_10_ and V_1_ that arise from ROS formation can be evaluated. V_10_, when administered in vivo, produces a subcellular distribution of vanadium in mitochondria that differs from that observed after exposure to vanadate [29]. Furthermore, a distinct pattern of LPO and markers of oxidative stress was observed in response to V_10_ different than that induced by V_1_ (see below) [28]. Neither V_10_ nor V_1_ affected the production of ROS, specifically superoxide anion (O_2_**.**^−^), in rat liver mitochondria except when NADH was used. V_10_ decreased O_2_**.**^−^ production in hepatic mitochondria by 40% (5 µM), whereas a 10-fold higher concentration of V_1_ was required to promote similar inhibition [27]. In cardiac mitochondria from fish, V_10_ had more potent antioxidant activity than V_1_, inhibiting the production of O_2_**.**^−^ in the absence (IC_50_ = 610 nM) and in the presence of NADH (IC_50_ = 15 nM) [28]. In the presence of NADH, V_1_ caused inhibition of O_2_**.**^−^ formation on the order of 50% (237 nM) [122,123].

This decrease in O_2_**.**^−^ formation was due, at least in part, to the inhibition of mitochondrial respiration. O_2_.^−^ arising from mitochondrial respiration is a stoichiometric precursor of mitochondrial H_2_O_2_. Even a small decrease in the membrane potential of rat brain mitochondria, which in turn decreases mitochondrial respiration, strongly inhibits ROS formation [136], which depends on the mitochondrial membrane potential. Since the V_10_ vanadate strongly depolarizes mitochondria [122,123], it should also decrease O_2_**.**^−^ production, which, in fact, has been observed in V_10_-treated isolated mitochondria.

LPO occurs after the induction of ROS production in the cell medium. There were several studies that, without specifying the species of vanadate present, have reported that vanadate can attenuate lipid peroxidation in hepatic tissue affected by hepatocarcinogenesis [137] and mammary carcinogenesis [138] without altering lipid peroxidation in control animals. Aureliano’s research group has demonstrated that a high concentration of vanadate (5 mM total vanadium) significantly increased LPO propagation in cardiac tissue 1 and 7 days after intravenous (i.v.) exposure to V_10_ (Figure 12B) [21]. In hepatic tissue, V_10_ had almost no effect on LPO 12 h after administration but by 24 h increased LPO to levels similar to metavanadate (Figure 12C) [27].

Whole heart tissues were similarly studied following in vivo intravenous (i.v.) administration using shorter exposure times (1, 6 and 12 h) and lower vanadium concentrations (1 mM) (Figure 12A). After 1 h, both V_10_ and V_1_ increased (*p* < 0.05) the baseline value, 3.02 ± 0.51 μmol TBARS/g tissue, by about 20% [28]. Only V_1_ sustained this increase in LPO at 6 and 12 h. V_10_ treatment had no prolonged effect. In a different species (*H. didactylus*), a higher vanadate concentration (5 mM total vanadium) with intraperitoneal administration and longer exposure times caused an increase in the propagation of LPO in cardiac tissue 7 days after administration (about 80% and 60% after treatment with V_10_ and V_1_, respectively) [26]. Together, these results suggest that V_10_ induces peroxidation through a different mechanism or perhaps even prevents V_1_ effects since it never achieves levels as high as those recorded 6 and 12 h after exposure to the V_1_ solution. Even with time, the total decomposition of V_10_ into V_1_ does not cause the same effects as those seen following the administration of V_1_. This suggests that interactions promoted by labile oligomeric oxovanadates are different from those induced by V_10_. Furthermore, when the V_10_ is completely decomposed into other oligomers of vanadate, targets may no longer be available that produce the effects noted previously. Still, at higher concentrations, V_10_ is unequivocally a promoter of LPO since V_1_ has less pronounced effects. Peroxidative damage could be related to a decrease in the activity of antioxidant enzymes as described in complementary studies following changes in antioxidant enzymes [21,27,29,71,122,123]. Although in most vanadium toxicity studies, the contribution of V_10_ is often not considered, it appears that, due to its longer stability at physiological pH, V_10_ does not completely decompose to V_1_ before causing marked changes in oxidative stress markers in vitro and in vivo. In fact, the V_10_ solution exhibits different patterns of LPO response and markers of oxidative stress than those resulting from V_1_ exposure [19,25,26,27,28].

In vivo studies underscore the fact that V_10_ induces consequential responses by antioxidant enzymes and LPO which appear to be related to observed increases in intracellular vanadium [19,25,26,27,28]. After administration of the V_10_ solution, metabolism was affected, with mitochondria subsequently identified as the main subcellular target. Thus, based on the in vivo results described here, it appears that V_10_ exhibits pro-oxidant activity by promoting the formation of superoxide anion (O_2_**.**^−^). The increase in radical production leads to a concomitant increase in superoxide dismutase (SOD) activity and an increase in H_2_O_2_. Increased H_2_O_2_ causes increased activity of glutathione peroxidases (GPx) which leads to an increase in glutathione (GSH) content. O_2_**.**^−^ also promotes the propagation of LPO through Haber–Weiss chemistry (Figure 13).

Importantly, the oxidative activity of V_10_ does not result directly from the induction of H_2_O_2_ production since the antioxidant activity of CAT appears to be unrelated to responses to oxidative stress induced by the toxic effects of V_10_. This suggests that the peroxidation of membrane lipids recorded after in vivo administration of V_10_ does not result from a stimulation of Fenton reactions. Results obtained after in vivo exposure to V_10_ differ from the effects of V_1_ and underscore the importance of taking into account the speciation of vanadium when evaluating its toxicity.

### 3.4. The Effect of LPO on DNA and Apoptosis

Vanadium affects DNA through multiple mechanisms. Vanadium can have direct interactions with DNA [139,140,141,142] or with proteins associated with DNA (histones), as well as indirect consequences through the generation of ROS. The direct interaction of vanadium complexes with DNA often reflects the ability of vanadium compounds to be intercalated into the 3D structure of the DNA double helix [143,144,145,146,147]. Vanadium salts and compounds have been shown to interact directly with proteins as discussed in Section 2.6. ROS are known to react with DNA as well [4,101,148]. Since vanadium enhances ROS formation, vanadium salts and complexes are also able to interact indirectly with DNA by first forming ROS that react in turn with DNA [4,118,149].

The main site of ROS interaction with DNA is the guanine base, where the product of the oxidation reaction is 8-hydroxyguanine. However, other sites of DNA interaction have been identified, including, for example, 2-deoxyribose. These reactions all cause changes in the basic DNA structure corresponding to mutations and potentially leading to programmed cell death and apoptosis. Both experimental studies and theories have emerged to explain the effect that ROS have on DNA and, as a consequence, on programmed cell death [4,118,148,150]. Since ROS are linked to LPO, the direct interaction of ROS with DNA must be considered as well.

Because ROS are increased by vanadium, they should be considered when evaluating vanadium effects on LPO while also recognizing that the direct effects of vanadium may be toxic as well. Some studies have found that vanadium, in the form of vanadate, can cause damage to DNA through DNA scission, phosphorylation inhibition and oxidation of DNA bases [151,152,153]. Individuals exposed to vanadium pentoxide through inhalation had greater amounts of DNA base oxidation [151]. In addition, DNA tail strands were lengthened due to increased purine oxidation (7%) and pyrimidine oxidation (30%). Desaulniers and coworkers found that vanadium in the form of sodium metavanadate decreased DNA phosphorylation and changed DNA unwinding due to breaks in the DNA strands [152]. Rodriguez-Mercado and coworkers found that V^IV^ in vanadium tetraoxide (V_2_O_4_) caused double-strand DNA breaks in human leukocytes, although V^III^, V^IV^ and V^V^ caused some DNA damage as well [153].

Damage to the DNA can lead to cell damage and death [154,155]. Similarly, ROS can play a role in apoptosis [156]. There are a number of pathways that lead to apoptosis that are affected by ROS and, in some cases, by LPO. Two pathways known to be affected by vanadium-generated ROS are the ERK/MAPK and the NF-kB pathways [33,157]. Vanadium inhibits the NF-kB pathway which causes cell proliferation in healthy cells and apoptosis in cancer cells. Molinuevo et al. found that two vanadyl compounds were related to the activation of the ERK pathway and enhanced apoptosis through a mechanism that was unclear, although treatment with antioxidants inhibited the pathway [157]. The effect on proteins (p53 and Cdc25B_2_) and on mitochondrial membrane permeability by vanadium also contributed to cell death [33,119,158]. These toxicological effects on otherwise healthy intact cells may also lead to cell death through effects on cell growth checkpoints [159].

### 3.5. The Oxidative Damage by V_10_ in Biological Systems

Several biological studies have shown that vanadium has the ability to produce ROS resulting in LPO and changes in antioxidant enzymes indicative of oxidative stress [40,65,66]. Despite accumulated knowledge to date, the oxidative effects of vanadium observed after acute exposure in vivo in cardiac muscle have not been thoroughly investigated, and the contribution of different oligomeric species to the toxic effects promoted by V^V^ is not known. In fact, most studies on the toxicological effects of vanadate in biological systems do not account for the contributions of different vanadium species even when the stable V_10_ is being used. Moreover, V_10_ can form intracellularly [62] and affect the activity of several enzymes [22,23,24]. The decomposition of V_10_ is sufficiently slow to allow the study of its effects on biochemical systems not only in vitro but also in vivo [20,25,26,27,28,29,71,122,123,160]. Furthermore, V_10_ interacts with certain cytoskeletal proteins such as actin [83,84,161] and with membrane proteins such as the Ca^2+^-ATPases [85,119,120] which increase its stability [84]. Interactions with other proteins such as myosin and with membrane vesicles do not affect the half-life time of V_10_ decomposition [83,162]. Therefore, it is hypothesized that V_10_ exists under physiological conditions for a period of time sufficient to induce several biological effects that differ from those promoted by V_1_. Careful evaluation of this possibility would require experimental conditions where V_10_ is stable such as a lower pH [160] or where V_10_ is associated with a protein [24]. Open questions are whether V_10_ exists under physiological conditions and, if so, whether this species induces specific in vivo effects such as LPO. In fact, a limited number of in vivo studies have been carried out with V_10_. Since 1999, Aureliano’s research group and its collaborators have carried out in vivo studies administering V_10_ solutions in an attempt to assess the contribution of the V_10_ to the toxic effects of vanadate [20,25,26,27,28,29,71,122,123,160,163], and more recently, Trevino and collaborators have investigated the therapeutic potential of V_10_ [80,164].

## 4. Vanadium Effects on Lipid Peroxidation and Disease Processes

The number of metal-based therapeutics is steadily increasing as the chemical space continues to be expanded. Vanadium is one of the less utilized elements to date with fewer ongoing clinical trials [54]. However, as dogma continues to be challenged and new discoveries are reported, this element is becoming more attractive to the pharmaceutical industry. Vanadium is, in general, known to induce oxidative stress. Although vanadium compounds can impact LPO, few investigations have been reported using compounds other than salts. Since ligands allow for the fine-tuning of the properties of vanadium, it is surprising that LPO studies with more compounds have not been undertaken.

### 4.1. Role of Vanadium in Lipid Peroxidation Related to Cancer

Despite somewhat controversial reports suggesting that vanadium may be an essential trace element for humans [165,166], pharmacological amounts of vanadium needed to observe efficacy may be 10 to 100 times the normal intake [167]. Under certain levels, some vanadium complexes/compounds have shown anticancer and/or antidiabetic activity in mammals [23,53,54,74,168], while higher levels can cause toxicity. Under some conditions, vanadium can act as a pro-oxidant molecule, which interacts with other oxidants and synergistically enhances oxidative stress and potentially lipid peroxidation (LPO) [11].

Reports from the early 1990s showed that V complexes/compounds induced LPO, which was associated with tissue toxicity and carcinogenicity [169,170,171]. Tissue-specific responses were shown for vanadate, which produced a cytotoxic response in the murine osteoblast-like MC3T3E1 nontransformed cell line [172]. This level of cytotoxicity was higher than that in vanadate-treated osteosarcoma cancer UMR106 cells with respect to both time- and concentration-dependent responses [172]. Osteoblastic cells were more sensitive to the vanadate-induced free radical and biomarker thiobarbituric acid (TBARS) formation, particularly at low concentrations. Nevertheless, higher basal TBARS was observed in untreated osteosarcoma cells [172]. Other vanadium compounds (VOSO_4_) and a complex of vanadyl with aspirin (VO/Aspi)) were found to be more potent than vanadate in inducing TBARS and inhibiting the cellular growth, in both cell lines tested [172] (Table 2). However, when an equivalent low concentration of VO/Aspi was released from a controlled delivery system (poly(β-propiolactone) (PβPL) film), less TBARS formation was observed [173] (Table 2), which reflects lower cytotoxicity compared to that previously reported for the metallodrug in solution [172].

The development and testing of vanadium derivatives with different ligands and with improved bioavailability and toxicity profiles continues. Both naproxen- and glucose-complexed vanadium compounds (NapVO and GluVO) had antiproliferative effects that were more pronounced in osteosarcoma UMR106 cells than in the normal MC3T3E1 osteoblasts [157]. This supported the observation that a low level of GluVO and NapVO increased TBARS production in tumoral cells but not in the nontransformed cells [157] (Table 2), suggesting LPO was involved in the antineoplastic action observed. Interestingly, neither the free vanadyl cation nor ligands induced an antimitogenic effect in cells at the concentrations tested [157]. At low concentrations, a large number of different complexes/compounds of vanadium were found to be therapeutically active [64,166]. Possible mechanisms for the anticancer activity of vanadium complexes/compounds included an increase in ROS generation, hyperactivation of the Ras-Raf-MEK-ERK pathway and cell cycle arrest [174,175,176]. It is also possible that vanadium may confer protection against chemical-induced carcinogenesis or toxicity in normal tissues by normalization of increased pathogenic LPO and oxidative stress. While an increase in hepatic LPO was observed in a group of carcinogen-treated female Sprague Dawley rats, this increase was lowered towards normal values by vanadium co-administration [138] (Table 2) and was associated with a significantly lower percentage of rats with tumors after vanadium treatment. In these experimental groups, SOD activity in the liver paralleled LPO. By contrast, hepatic glutathione (GSH) and cytochrome P450 (CYP) enzyme content and glutathione S-transferase (GST) activity decreased with carcinogenic treatment compared to control rats and recovered with vanadium treatment [138]. Similarly, in a model of hepatocarcinogenesis induced in rats by chronic feeding of 2-acetylaminofluorene (2-AAF), continuous vanadium administration inhibited LPO and suppressed cell proliferation [177] (Table 2), suggesting vanadium was chemopreventive.

The chemoprotective role of vanadium against cancer chemotherapy-induced toxicity is also relevant. Many chemotherapeutic agents such as cyclophosphamide (CP) and cisplatin (CDDP) are toxic due to multifactorial mechanisms that include increased oxidative stress in normal tissues and organs, namely the liver and kidney. The co-administration of compounds with antioxidant potential may be beneficial to patients. For example, the simultaneous treatment of female Swiss albino mice with CP and either vanadium(III)-L-cysteine complex (VC-III) [178] or oxovanadium(IV)-L-cysteine methyl ester (VC-IV) [179] reduced ROS levels when compared to the increase in ROS observed in CP-treated group vs. control [178,179]. With respect to LPO, partial normalization of TBARS in CP/VC-III- or CP/VC-IV-treated mice was observed (Table 2) [178,179]. After treatment with CP, there was a decrease in GSH levels and in GST, glutathione peroxidase (GPx), superoxide dismutase (SOD) and catalase (CAT) activities, while a recovery was observed with vanadium treatment [178,179]. Similar protective effects were observed with concomitant treatment with cisplatin (CDDP) and VC-III (Table 2) [180]. These results suggest that vanadium may be beneficial as an adjunct therapy to protect against the toxicity of anticancer drugs.

**Table 2 ijms-24-05382-t002:** Effects of vanadium in lipid peroxidation related to cancer. Main outcomes of studies using different V compounds in various organs/tissues of animal models or cancer cells.

Vanadium Compound	Combined/Complexed	Carcinogenic/Toxic Agent or Cell Lines	Tissue/Model	Main Results/Outcome	Ref.
V^1V^; VO	Aspirin; polymeric film	Osteosarcoma UMR106 cells in culture	Bone	Cytotoxic effects	[172,173]
V^1V^ derivatives	Naproxen (Nap-VO); Glucose (GluVO)			Apoptosis mediated by lipid peroxidation	[157]
Ammonium monovanadate (NH_4_VO_3_, +V oxidation state) (vanadium supplemented in drinking water)		7,12-dimethylbenz(a)anthracene (DMBA)-induced mammary carcinogenesis in rats	Mammary gland	Prevention of mammary cancer	[138]
Vanadium (in the form of ammonium vanadate)		2-acetylaminofluorene (2-AAF)-induced hepatocarcinogenesis in rats	Liver	Vanadium was chemopreventive; inhibition of lipid peroxidation	[177]
Oxovanadium(IV)-L-cysteine methyl ester (VC-IV)		Cyclophosphamide (CP)-induced hepatotoxicity in mice	Liver	Protective role of VC-IV against CP-induced toxicity	[179]
Vanadium(III)-L-cysteine complex (VC-III)				Protective role of VC-III against CP- and CDDP-induced toxicity	[178]
		Cisplatin (CDDP)-induced nephrotoxicity in mice	Kidney		[180]

Even though vanadium participates in Fenton-type reactions [40] and the mechanisms proposed for vanadate action involve redox cycling and the production of ROS [65,66], some results show a depression in ROS and the rate of ROS formation [21]. Previous results show that in certain experimental conditions, for example in rats with induced hepatocarcinogenesis [137] and diabetes [138], vanadate may decrease oxidative stress. Strong evidence supports the observation that V_10_ alters the production of mitochondrial O_2_.^−^ differently from V_1_ and suggests the possibility that different pathways are involved in the biological activity of different vanadium species. Of the proposed intracellular pathways for vanadate, several involve the production of O_2_.^−^ mediated by oxidoreductases of NADPH in the respiratory chain [65,66]. Considering the proposed mechanisms of action and detoxification of vanadate, which include reducing vanadate to vanadyl with O_2_**.**^−^ production, the available data support the interpretation that V_10_ may participate in bioprocessing and metabolism differently than V_1_.

### 4.2. Effect of Vanadium in Diabetes-Induced Lipid Peroxidation

Diabetes mellitus is a complex metabolic disease characterized by a chronic state of hyperglycemia [181]. Although the impaired function of the pancreatic islets might be relevant in its etiology, other tissues are affected and may present complications in uncontrolled disease [182]. Diabetes can be generally classified into different categories with distinct clinical features. Type 1 diabetes is an autoimmune disease in which beta cells in the pancreas are unable to produce the hormone insulin while in type 2 diabetes, the most common form of diabetes, the body is either resistant to insulin or incapable of producing sufficient amounts of insulin [181].

An imbalance between the production and removal of ROS and RNS may contribute to insulin resistance and pancreatic beta cell dysfunction, which ultimately leads to the development of type 2 diabetes [183,184]. Increased levels of TBARS, a biomarker of LPO, were more highly elevated in type 2 diabetes patients than in healthy control subjects [185]. In patients with type 2 diabetes, hyperglycemia was associated with increased oxidative stress and free radical-mediated LPO [186,187,188] both of which may affect the development of micro- and macrovascular complications related to the intensification of systemic inflammation in these patients [184,189,190]. Compounds that modulate LPO and oxidative stress and have an antioxidant potential may contribute to improving the metabolic health in patients with diabetes and be a valuable therapeutic approach.

In 1979, Tolman et al. showed that vanadium salts exhibited insulin-mimetic effects which led to an interest in vanadium chemistry for the treatment of diabetes [191]. Since then, a series of reports have been published describing the insulin-like effects of various vanadium compounds, mainly V^IV^ and V^V^ salt and coordination complexes. One coordination complex, an organic vanadium compound, bis(ethylmaltolato)oxovanadium(IV) (BEOV), exhibited excellent efficacy in streptozotocin (STZ)-diabetic rats [192] and entered Phase I and II clinical trials [193,194].

Using different animal models of diabetes and analyzing diverse tissues, many reports showed effects of vanadium compounds on the activity of antioxidant enzymes and the levels of LPO (Figure 14). Early studies from the 1990s showed that treatment of STZ-induced diabetic Sprague Dawley rats with sodium metavanadate (NaVO_3_), did not lead to changes in the antioxidant defense system [195]. However, the tissue level of vanadium positively correlated with the TBARS level [195]. By contrast, sodium orthovanadate (SOV) treatment of STZ-induced diabetic male Wistar rats led to the STZ-induced decrease in the hepatic activities of SOD, CAT and GPx being restored to normal levels, while the elevated plasma lipid peroxides (as measured by MDA) were decreased almost to basal values [196] (Figure 14). This same pattern was observed in the liver enzymes of alloxan-induced diabetes female Wistar rats (Figure 14), but not in all the tissues evaluated [137]. SOV treatment also almost normalized the chemical-induced increase in the levels of TBARS in the brain, along with the normalization of the activity of the brain GST, which was decreased in the diabetic rats [197] (Figure 14). Interestingly, subsequent studies have used SOV combined with *Trigonella graecum* seed powder (TSP) which makes it possible to use lower concentrations of vanadate. Most authors have shown a reversal of non-physiologic antioxidant levels and peroxidative stress in different tissues from diabetic animals [198,199,200,201,202] (Figure 14).

The vanadium salt in oxidation state IV, vanadyl sulfate (VOSO_4_), has also been studied extensively. An early study showed that TBARS levels were elevated in vanadyl-treated animals, although cataract development was suppressed in STZ-diabetic Wistar rats [215]. By contrast, many reports showed in different tissues that the treatment with VOSO_4_ reversed the increased levels of LPO in response to diabetes induction [205,206,207,208,211,212] (Figure 14). These results were recently expanded to show similar normalization of the oxidative state in cardiac, lung, skeletal muscle and eye lens tissue [216]. Similar antioxidant effects were observed in the pancreas, liver and kidneys of diabetic rats treated with a macrocyclic binuclear oxovanadium complex (MBOV) [213,214] and in the pancreas and brain of diabetic rats treated with the N(1)-2,4-dihydroxybenzylidene-N(4)-2-hydroxybenzylidene-S-methyl-thiosemicarbazidato-oxovanadium (IV) (VOL) compound [209,210] (Figure 14).

Oxidative stress in the liver and muscle tissues of alloxan-induced diabetic rats was addressed after treatment with (H_2_Metf)_3_[V_10_O_28_] (metformin-decavanadate, MV_10_) [204]. After 60 days, decreased activity levels of SOD and CAT induced by alloxan were restored to normal levels (Figure 14). Furthermore, the increased levels of LPO markers in the diabetic animals were normalized after Metf-V_10_ treatment. This was observed for both MDA and 4-hydroxyalkenal (4HDA) levels in a similar fashion to treatment with insulin, while metformin alone had very limited effects [204] (Figure 14). Decavanadate was previously reported to increase the glucose uptake in rat adipocytes, in the presence or in the absence of insulin [217]. Together, these findings suggest that vanadium compounds are not only insulin mimetics but may also enhance the activity of insulin [23,164,218].

Changing the oxidation state of the vanadium compound changes the redox properties of the complex alters the formation of LPO products as described above for vanadate (V^V^) and VOSO_4_ (V^IV^). One study compared the effects of vanadium in oxidation states III, IV and V in a series of coordination complexes with the same ligand, chloro-substituted dipicolinic acid [203]. V^IV^dipic-Cl and V^V^dipic-Cl complexes in liver tissues produced improved blood glucose levels, while there were lesser effects of V^III^dipic-Cl [203] (Figure 14). This demonstrated that even high-oxidation-state vanadium compounds are beneficial in changing MDA levels toward normal and reducing ROS levels presumably through redox cycling. For complexes with the dipic-Cl ligand, it was surprising that the V^V^ complex showed a trend towards being slightly better at normalizing the redox state of diabetic cells, though V^V^ complexes would need to undergo Fenton chemistry first [203].

It is important to note that the animal models of diabetes using diabetogenic chemicals cause the destruction of β-cells resulting in type 1 diabetes, so it is unclear whether such effects would be observed in type 2 diabetes animal models or patients. Moreover, in some cases, the diabetic animals did not show a reduction in the activity of enzymes involved in antioxidant defense in all tissues analyzed. As an example, an increase in enzyme activity was observed in diabetic heart tissues when compared to normal animals [137,201], although even in such cases, treatment with vanadium compounds restored levels close to normal (non-diabetic) values [137,198,201,206].

Taken together, these data suggest treatment with vanadium compounds may contribute to alleviating oxidative stress in patients with diabetes and contribute to an overall improvement in metabolic function. However, more evidence of vanadium antioxidant beneficial effects and safety is still required.

### 4.3. Vanadium Lipid Peroxidation and Neurodegenerative Diseases

Vanadium is known to have neurotoxic effects and contribute to a number of neurodegenerative diseases presumably through the introduction of oxidative stress and LPO production. The brain contains high amounts of PUFAs, making it a prime target for LPO, which can cause the destruction of the myelin sheath, loss of neurons via cell death, disruption of the cell membrane potential, depletion of dopamine, and inactivation of phosphatase enzymes. Neurons are surrounded by a myelin sheath which is important for the development of the electric potential and the ability to transmit electrical impulses in the form of action potentials quickly. Vanadium exposure has been reported to cause damage to the myelin sheath [219] and, as a result of LPO, neuronal death. LPO in the mitochondria also leads to cell death through effects on mitochondrial membranes. Vanadium accumulates in the brain after exposure [220], indicating that the toxic effects of vanadium relating to membrane destruction may play a role in the reported neurodegenerative diseases such as Parkinson’s and Alzheimer’s. The metal content and transporters in the rat brain have been reported to be sensitive to the presence of other metals, including Mn, chromium, zinc, cobalt, aluminum, molybdenum and vanadium [221].

#### 4.3.1. Parkinson’s Disease

Parkinson’s disease (PD) is a neurodegenerative disease that has been associated with several failures in brain function. A decrease in the neurotransmitter dopamine has been correlated with the onset of Parkinson’s which, with disease progression, leads to a failure in the dopaminergic system. Some basis of knowledge around metals, specifically manganese (Mn), and the onset of Parkinson’s or the onset of similar symptoms called Parkinsonism exists [222]. The latter is a condition that results in loss of motor and neurological function similar to that of Parkinson’s but does not exhibit the symptoms of Parkinson’s disease. Symptoms produced by Mn are called manganism [223]. Mn, like vanadium, undergoes redox cycling and is known to have many neurotoxic effects. The ability of Mn to produce ROS has been well characterized [224] and shown to cause effects on mitochondrial function similar to those observed with vanadium treatment, including the loss of the mitochondrial membrane potential and the release of Cyt C [225]. Additionally, LPO products have been observed in mitochondria and the endoplasmic reticulum system and are similar to the oxidative stress in response to manganese exposure. With high doses of Mn, symptoms of Parkinson’s disease are seen and correlated with the onset of Parkinson’s [226]. Given the similarities between vanadium and manganese, the effects of vanadium on the onset of Parkinson’s are likely to be similar. Ngwa and coworkers have reported a link between vanadium neurotoxicity and its effect on the dopaminergic system due to its effect on protein kinase C-delta and its function in cell signaling mechanisms [227]. Ohiomokhare and coworkers (2020) found that vanadium increased ROS and decreased motor function in Melanogaster drosophila, both wild-type and PD models, and that these effects were alleviated with chelators or the administration of L-DOPA [228].

#### 4.3.2. Alzheimer’s Disease

Alzheimer’s disease (AD’s) is a neurodegenerative disease characterized by loss of memory. Although no single cause of AD’s has been discovered, there is evidence that metals, lipid peroxidation and oxidative stress can play a role in disease progression. The disease is associated with the accumulation of β-amyloid plaques in the brain that have the capability of interacting with redox-active metals, such as copper, zinc and iron [229]. These metal ions induce the disease because of their ability to generate ROS and damage the brain through DNA damage and oxidation of lipids and proteins. Studies have shown that 4-HNE, a product of LPO, is present in the brains of AD’s patients [230]. Mitochondrial ROS production and mitochondrial dysfunction have been associated with AD [231]. All of these are known products of vanadium-based oxidative stress and offer a basis for vanadium having a role in AD.

As is the case for Parkinson’s, there are only a limited number of studies characterizing vanadium effects on the development of AD’s disease or its progression. However, due to vanadium’s redox properties and ability to generate ROS, it has the potential to induce at least some similar effects. Montiel-Flores and coworkers found that the inhalation of vanadium pentoxide caused AD-like neuronal cell death in rats [232].

There is also a growing body of work investigating the use of vanadium in treating AD’s disease. Although vanadium has toxic effects, studies reported some potential of vanadium-based therapeutics for AD’s disease [233]. Vanadyl acetylacetonate was found to promote glucose and energy metabolism in neuronal cells but did not reduce β-amyloid plaque production [234]. Two peroxovanadium complexes were reported to inhibit β-amyloid fibril formation. He et al. (2015) showed that two complexes were able to inhibit the aggregation of amyloids using PrP106–126 and Aβ_1–42_, where PrP is from the prion disease and refers to protein-prion protein. Inhibition was more effective in PrP than in Aβ, but there was not much difference in its effects on cell toxicity. Peroxovanadium complexes increased cell viability perhaps due to the ability of peroxovanadium complexes to reduce methionine residues [235]. This group also found that BEOV was able to ameliorate AD symptoms through a number of mechanisms including inhibition of Aβ aggregate formation [236]. These results should encourage studies on the use of vanadium in the treatment of neurodegenerative diseases.

### 4.4. The Potential for LPO as a Future Target for Therapeutic Treatments

The ability of vanadium compounds to impact oxidative stress and the formation of LPO products is well documented [237]. Since vanadium remains a comparatively underexplored metal [238], new compounds are being assayed to determine their potential for alleviating oxidative stress [239,240]. Novel compounds are being designed which affect LPO but lack cellular toxicity. New pathways are discovered by investigating organisms not traditionally investigated [241]. New approaches are being developed based on combatting oxidative stress in disease processes. For example, a 2D vanadium carbide synthetic enzyme referred to as V2C MXenzyme has been reported to alleviate ROS-mediated inflammation [224]. Specifically, the 2D V2C MXenzyme can replace SOD, CAT, POD, TPx, GPx and HPO, thus mimicking the intracellular antioxidant defense system against ROS-mediated oxidative damage including protein carbonylation, lipid peroxidation and DNA damage. In vitro and in vivo experiments demonstrated that V2C MXenzyme was biocompatible and exhibited ROS-scavenging capability, protecting cellular components against oxidative stress. Future investigations are likely to involve the characterization of novel biological systems, new compounds and agents such as the V2C MXenzyme and related systems [224] designed to combat the effects on oxidative stress and LPO.

## 5. Conclusions

Lipid peroxidation (LPO) is a process that affects human health and can be modulated by vanadium compounds. Some forms of vanadium salt and complexes exacerbate LPO, while other compounds have protective effects. Vanadium salts and compounds can sometimes generate radicals and ROS directly but can also act indirectly through effects on LPO. LPO typically affects the structure and function of cellular membranes, as do vanadium compounds. The formation of ROS also impacts other cellular functions in ways that are modulated by vanadium compounds. The protective effects of vanadium compounds reported for cancer and neurodegenerative diseases are likely to involve LPO products and ROS.

The effects of vanadium compounds on several biological processes have been tested, and a number of biomarkers have been identified. The most common ones are malondialdehyde (MDA) and hydroxynonenal (4-HNE), although cis-parinaric acid and ascorbate have been used, as have enzymes such as superoxide dismutase (SOD) and catalase (CAT). The effects of LPO on enzymes, DNA and membrane uptake or signaling in the presence of vanadium compounds have been characterized. Enzymes with redox-active amino acids such as cysteine can be reduced in the presence of vanadium compounds, and hence the enzymes are irreversibly affected. Similarly, LPO can affect these enzymes. In the case of DNA, guanine is generally converted to 8-hydroxyguanine, and similarly, the direct interactions of vanadium impact the guanine base in DNA, although vanadium compounds can also be intercalated in the 3D-organized DNA strand.

Vanadium accumulation in cells suggests that mitochondria are frequently subcellular targets for vanadium, particularly when it is administered as V_10_. This conclusion is based on distribution studies of vanadium in the mitochondria of cardiac, hepatic and renal tissue. The extent of vanadium accumulation depends on the system investigated, the total concentration of vanadium administered and the mode of administration. Decavanadate (V_10_) is consistently 10 to 100 times more potent than monomeric vanadate (V_1_) both as a mitochondrial membrane-depolarizing agent and as an inhibitor of oxygen consumption by hepatic and cardiac mitochondria and may contribute to the antioxidant effect through the partial inhibition of ROS production. Although mitochondrial effects have generally been the target of vanadium studies, other cellular components and organelles should not be ignored.

Because vanadium salts and complexes can induce ROS formation both directly and indirectly, studies of mechanisms leading to LPO should include investigations of both direct and indirect processes. This is, of course, complicated by the fact that different vanadium species exist under physiological conditions and can have different effects. Thus, studies evaluating the mechanism of action of vanadium compounds should involve speciation studies of vanadium to evaluate the direct and indirect effects of the species that exist during the course of study. In addition, studies relating to how these molecules interact in other biological systems are relevant to these considerations. Undoubtedly, understanding LPO is important for better understanding how vanadium exerts its effects in biological systems and gives rise to some of the beneficial effects reported in cells and tissues in cancer, diabetes and neurodegenerative diseases, important therapeutic targets that will hopefully encourage future work with this important metal.

## Figures and Tables

**Figure 1 ijms-24-05382-f001:**
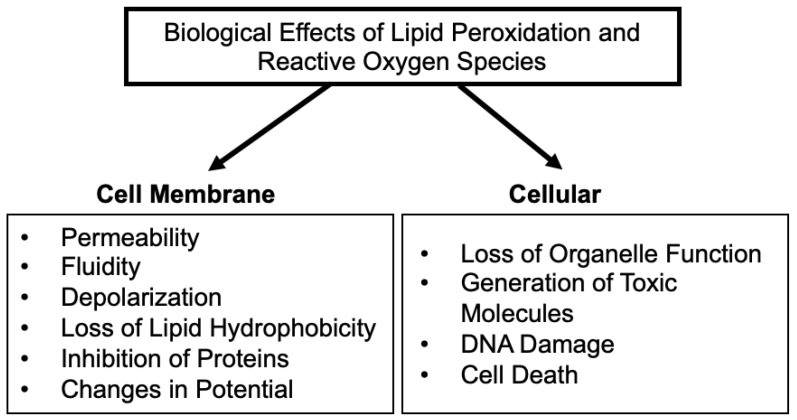
Overview of biological effects of lipid peroxidation (LPO) and reactive oxygen species (ROS). Adapted from [11] with copyright permission from Bentham.

**Figure 2 ijms-24-05382-f002:**
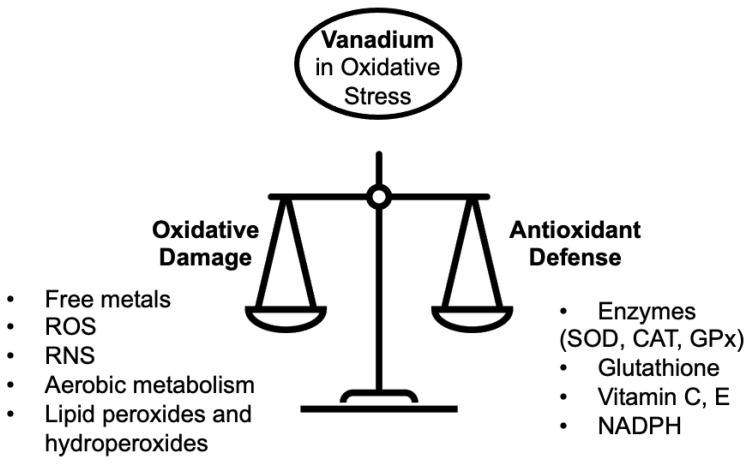
Molecules involved in oxidative stress with either unfavorable oxidative damage potential or the ability to provide antioxidant defense that is protective. Abbreviations: RNS, reactive nitrogen species; SOD, superoxide dismutase; CAT, catalase; GPx, glutathione peroxidase.

**Figure 3 ijms-24-05382-f003:**
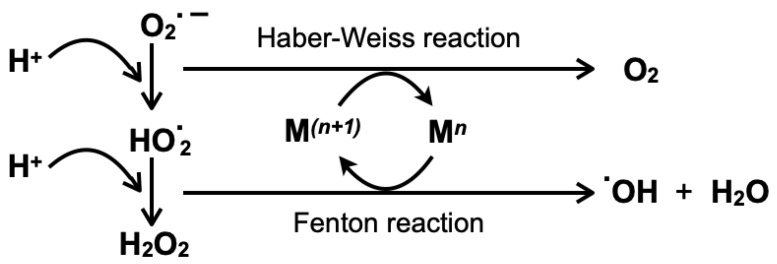
The relationship between the Fenton reaction and reactions catalyzed more generally by transition metals forming oxygen radicals. Metal-catalyzed radical formation is linked to the Haber–Weiss reaction as shown. Adapted with permission from [8]. Hindawi Copyright 2014, Antonio Ayala et al.

**Figure 4 ijms-24-05382-f004:**
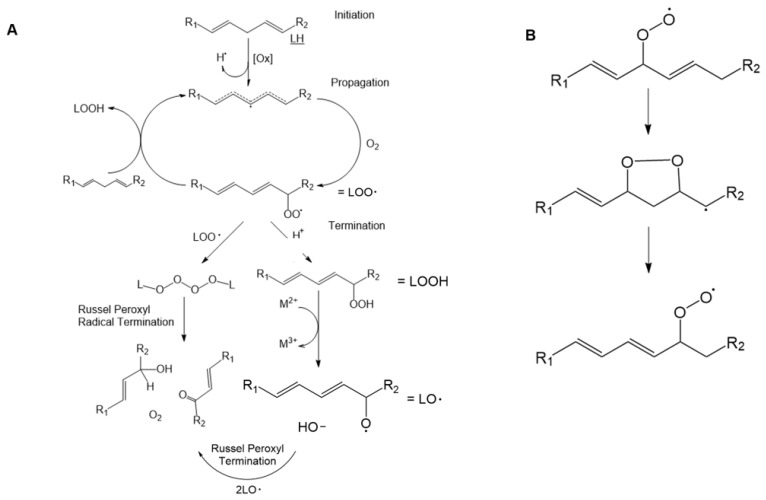
The overall mechanism of LPO. (**A**) Three phases of lipid peroxidation: initiation, propagation and termination, where LH = polyunsaturated fatty acid; LOOH = lipid hydroperoxide shown in scheme; LOO· = lipid peroxyl radical shown in scheme; LO· = lipid alkoxyl radical shown in scheme; [Ox] = an oxidant and could be L·, HO·, UV Light, HOO·; M = metal (Fe, V, Cu). (**B**) Peroxyl radical cyclization rearrangement, where R_1_ = C_5_H_11_; R_2_ = (CH_2_)_7_COOH.

**Figure 5 ijms-24-05382-f005:**
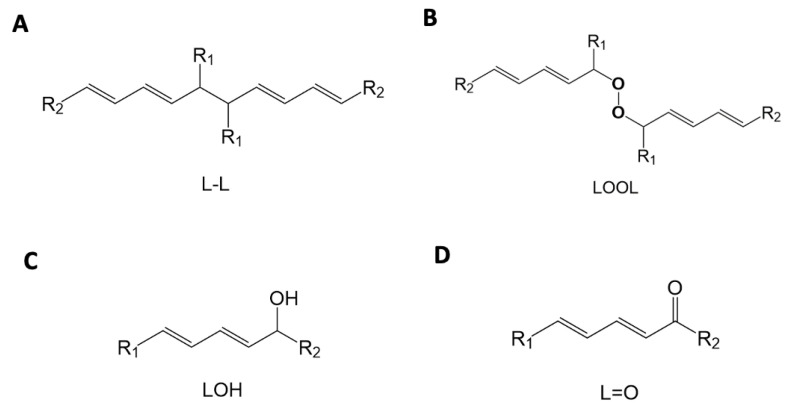
Core termination products shown in reactions (17), (18), (20) and (21) with R_1_ = C_5_H_11_; R_2_ = (CH_2_)_7_COOH. The stereochemistry of products through the following reaction steps are (**A**) self-termination, (**B**) cross-termination, (**C**) lipid hydroxyl and (**D**) lipid ketone.

**Figure 6 ijms-24-05382-f006:**
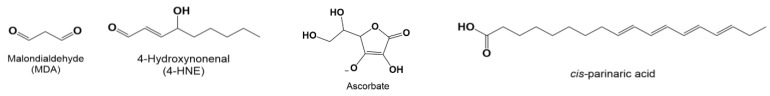
Structures, names and abbreviations of biomarkers for LPO products induced by vanadium salts or vanadium compounds including MDA, 4-HNE, ascorbate and *cis*-parinaric acid. MDA and 4-HNE are LPO products and excellent biomarkers. Ascorbate is an antioxidant, and *cis*-parinaric acid is a fluorescent probe sensitive to the presence of peroxyl radicals in cell media.

**Figure 7 ijms-24-05382-f007:**
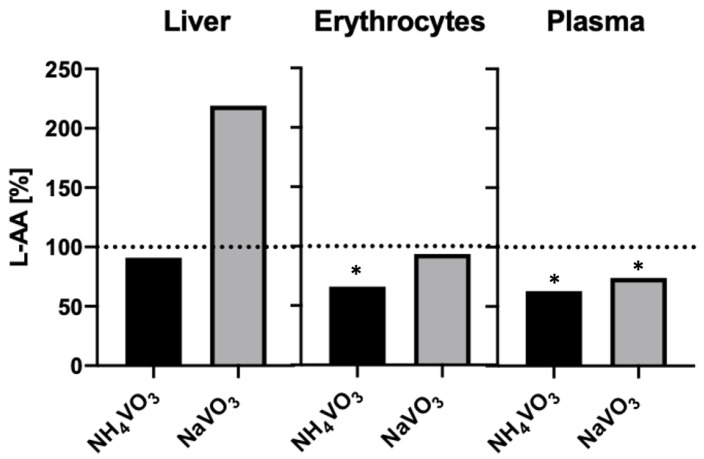
Levels of L-ascorbic acid (L-AA) in the liver, erythrocytes and plasma of rats receiving selected vanadate compounds expressed as the percent of changes compared to the control. The NH_4_VO_3_ dose (mg/body weight/24 h) was 10.7 mg for 4 weeks (wks) for all samples evaluated (liver, erythrocytes and plasma). The NaVO_3_ dose was 12–13 mg for 12 wks for liver and 10.7 mg for 6 wks for erythrocytes and plasma. L-AA = L-ascorbic acid; NH_4_VO_3_, ammonium metavanadate; NaVO_3_, sodium metavanadate. * designates statistical significance relative to control (*p* < 0.05). Adapted from [11] with copyright permission from Bentham.

**Figure 8 ijms-24-05382-f008:**
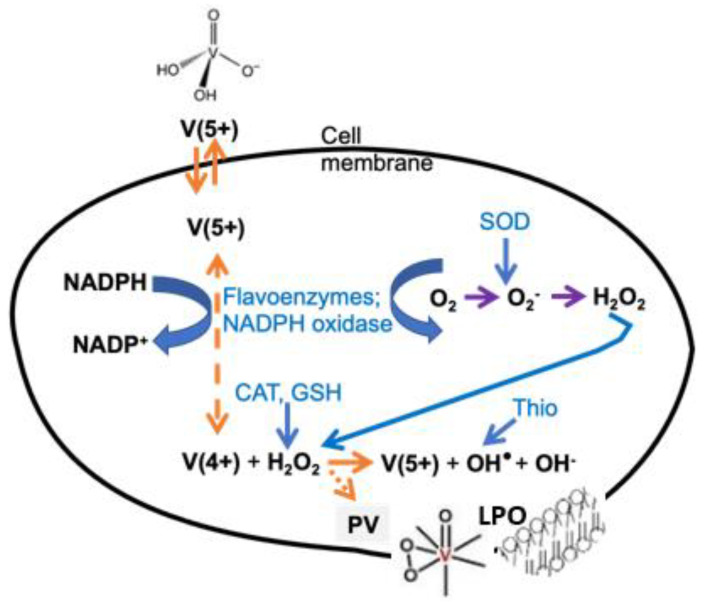
Vanadate (V(5+)), vanadyl (V(4+)) and peroxovanadate (PV) effects on intracellular oxidative stress. Representation of the mechanisms described in text for redox cycling under oxidation of NADPH, reaction with H_2_O_2_ undergoing oxidation and formation of vanadium peroxides that may interact directly with lipids. CAT, catalase; SOD, superoxide dismutase; PV, peroxovanadate; LPO, membrane lipid peroxidation. Adapted from [66] with copyright permission from Elsevier.

**Figure 9 ijms-24-05382-f009:**
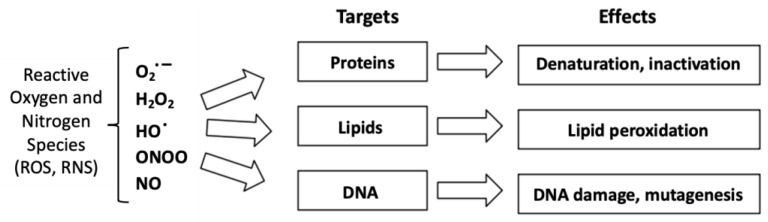
The consequences of reactions of reactive oxygen species (ROS) and reactive nitrogen species (RNS) with major biomolecules such as proteins, lipids and nucleic acids, promoting global structural modifications leading to denaturation and/or inactivation of proteins, lipid peroxidation, DNA damage and mutagenesis.

**Figure 10 ijms-24-05382-f010:**
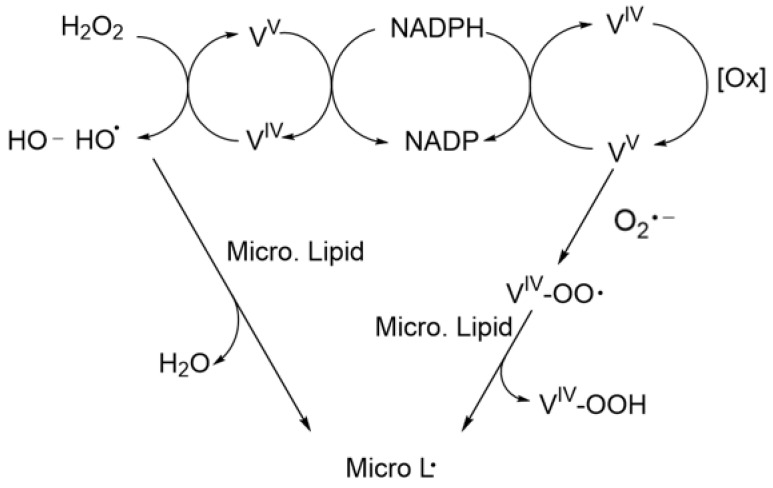
The recycling of V^V^ to V^IV^ catalyzed by H_2_O_2_ or NADPH is shown, as is the similar conversion of V^IV^ to V^V^ catalyzed by an oxidant or NADP. The proposed mechanism of vanadium redox cycling and conversion to LPO products (Micro L.) occurs in microsomes. Inspired by Ref. [112].

**Figure 11 ijms-24-05382-f011:**
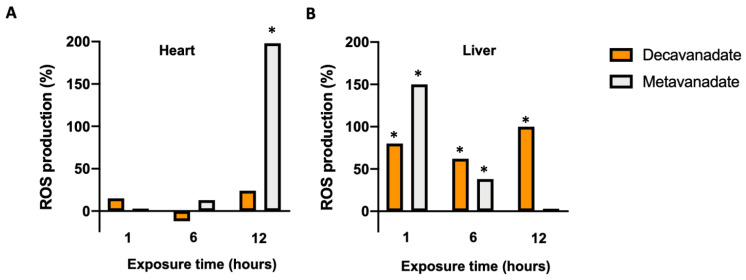
Percent ROS production is shown as a function of time in (**A**) cardiac and (**B**) hepatic mitochondria of *Sparus aurata*. Animals (n = 6) intravenously received solutions of V_10_ or metavanadate (1 mM total vanadium). * designates significance compared to control (*p* < 0.05). Adapted from [21,28] with Research Signpost and Elsevier copyright permission.

**Figure 12 ijms-24-05382-f012:**
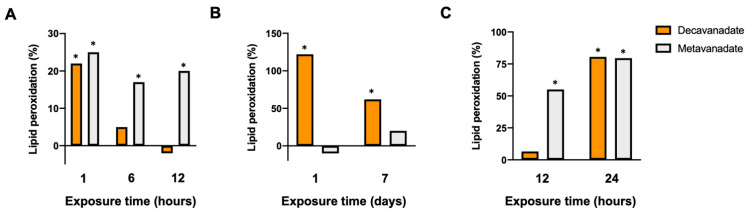
Lipid peroxidation products following in vivo administration of V_10_ and metavanadate (V_1_) as a function of time depended on the animal model used, the mode of administration, tissues being studied and vanadium concentrations. (**A**) Cardiac tissue (1 mM total vanadium), intravenous administration, *Sparus aurata*; (**B**) cardiac tissue (5 mM total vanadium), intraperitoneal administration, *Halobactracus didactylus*; (**C**) hepatic tissue (5 mM total vanadium), intravenous administration, *Halobactracus didactylus*. * designates significance compared to control (*p* < 0.05). Adapted from [21,27,28] with Research Signpost and Elsevier copyright permission.

**Figure 13 ijms-24-05382-f013:**
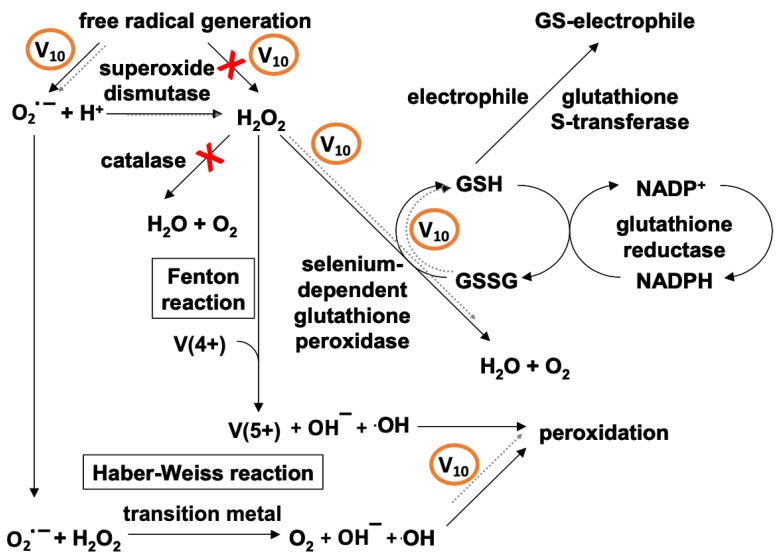
Putative V_10_ pathways for the generation of reactive oxygen species and of the actions in some of the enzymes involved in antioxidant defense mechanisms in cells. In vivo V_10_ studies point to the formation of the superoxide anion (O_2_.^−^), leading to increases in superoxide dismutase (SOD) activity, H_2_O_2_, the activity of glutathione peroxidases (GPx) and a subsequent increase in glutathione (GSH) content. Adapted from [21] with Research Signpost copyright permission.

**Figure 14 ijms-24-05382-f014:**
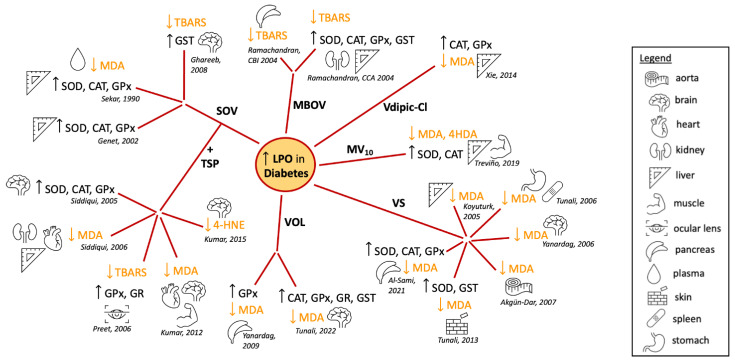
Summary of the reported effects of different vanadium compounds in lipid peroxidation and antioxidant enzyme activity, evaluated in different tissues of diabetes-induced animal models [137,196,197,198,199,200,201,202,203,204,205,206,207,208,209,210,211,212,213,214]. Abbreviations: superoxide dismutase, SOD; catalase, CAT; glutathione peroxidase, GPx; glutathione reductase, GR; glutathione S-transferase, GST; malondialdehyde, MDA; 4-hydroxy-2-nonenal, 4-HNE; thiobarbituric acid reactivity, TBARS; 4-hydroxyalkenals, 4HDA; sodium orthovanadate, SOV; Trigonella graecum seed powder, TSP; macrocyclic binuclear oxovanadium complex, MBOV; N(1)-2,4-dihydroxybenzylidene-N(4)-2-hydroxybenzylidene-S-methyl-thiosemicarbazidato-oxovanadium (IV), VOL; vanadyl sulphate, VSO_4_; metformin-decavanadate, MV_10_ [137,196,197,198,199,200,201,202,203,204,205,206,207,208,209,210,211,212,213,214].

**Table 1 ijms-24-05382-t001:** Potential steps and products formed in the termination reaction of lipid peroxidation.

Mechanism	Products
Russell peroxyl radical termination (4 oxygen)	Aldehyde/ketone, alcohol, molecular oxygen
Russell peroxyl radical termination (2 oxygen)	Aldehyde/ketone, alcohol, pentane
Antioxidant quenching	Lipid hydroperoxide, lipid alcohol
Self-termination	Crosslinked lipids
Cross-termination	LOOL
Metal-mediated	Hydroxyl radical, aldehyde/ketone, pentane

## Data Availability

The data are available in the original research papers.

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
