# Peer review of "Biological Consequences of Vanadium Effects on Formation of Reactive Oxygen Species and Lipid Peroxidation"

_ijms, 2023, doi:10.3390/ijms24065382_

Round 1

Reviewer 1 Report

The manuscript "Biological Consequences of Vanadium Effects on Formation of Reactive Oxygen Species and Lipid Peroxidation" addresses an important issue: Lipid peroxidation (LPO), a process that affects human health and can be induced by exposure to vanadium salts and compounds.

The objectives were clearly stated and explained in the manuscript, however the experimental strategy raises some major concerns and so the experimental information from which the conclusions were drawn. The manuscript is overall well written and has good organization. Quantitative analysis of the experimental data is missing throughout the manuscript and the interpretations of the results and the discussion are thus suffering from these limitations.

The paper is interesting but there is a need for more experimental detail in order to critically review the data. Specifically, they should provide information for the following questions and comments:

Major points:

1.      The authors should include more recent update on this topic and compare how this study further advances the current knowledge in the “Introduction section”.

2.      Caption in the majority of the figures is scarce and a more detailed description is needed, especially in Figures 6, 9, 10, 12, 13 and Table 2.

3.      Chemical symbols, chemical abbreviations and acronyms as well as their formal charges should be correctly spelled and written. Correctly use superscript when needed, e.g., in the reactions described in the text.

4.      Unify the style of the references in the References Section and add DOI in the cases it is possible. And use the same reference and citation (follow MDPI’s guidelines) style in the main text.

5.      The Methods section in the study should be more accurately described for each technique used.

6.      Statistical evaluation should be performed on the data shown in Figure 7, 11, 12 in order to better interpret the results and their significancy.

Minor points:

1.      Remove highlighted text in Figure 8. This Figure could also be increased in size for the sake of clarity.

Reviewer 2 Report

Reviewer comments and suggestions

The study discussed the biological consequences of vanadium effects on formation of reactive oxygen species and lipid peroxidation which is an important way to evaluate how vanadium exerts effects in biological systems concerning diabetic, neurodegenerative, and other diseased tissue The authors also suggested that speciation of vanadium, together with investigations of ROS and LPO, should be considered in future biological studies that were discussed in this review.

Below are the comments for this paper to be incorporated in the revised version of the manuscript. 

  1. Line 64-65 The sentence needs to be revised. 
  2. Line 79-80 The authors did not suggest the way they describe.
  3. Line 82-83 Please explain these studies
  4. After going through the manuscript, it needed an English language correction for removing grammatical errors.
  5. Line 170-173 why these references were repeated, is it necessary?
  6. In most of the figures the authors wrote adapted whether they have taken permission for this
  7. Typo error 636 line 
  8. Line 684 a number of studies require more references to be incorporated.
  9. LPO effects have been seen in diabetes, please include this aspect as well.
  10. All references need to be modified based on MDPI guidelines.

Round 2

Reviewer 1 Report

The authors have responded satisfactorily to the referees' comments and corrections.